# *Psyttala horrida* (Stål, 1865) (Hemiptera: Reduviidae: Reduviinae)—A Morphological Study of Eggs and Nymphs

**DOI:** 10.3390/insects13111014

**Published:** 2022-11-03

**Authors:** Agnieszka Bugaj-Nawrocka, Agata Danielczyk, Iga Sułkowska, Dominik Chłond

**Affiliations:** Institute of Biology, Biotechnology and Environmental Protection, Faculty of Natural Sciences, University of Silesia in Katowice, Bankowa 9, 40-007 Katowice, Poland

**Keywords:** assassin bugs, immature stages, scanning electron microscope, stages development, ultrastructure

## Abstract

**Simple Summary:**

We present a morphological and morphometric study of the eggs and all nymphal stages of *Psyttala horrida*—a representative of the predatory family Reduviidae. Research on eggs and nymphs is essential for individual groups of insects because they complement the knowledge about their development. They are also important information for systematic and phylogenetic research. For the first time, descriptions of all immature stages of *P. horrida* are presented, with the additional use of images from scanning electron microscopy (SEM).

**Abstract:**

In terms of body size, species of the genus *Psyttala* Stål, 1859, are the largest known representatives of the subfamily Reduviinae. Among the species belonging to this genus, *Psyttala horrida* (Stål, 1865) is the most popular, mainly because it is a laboratory breeding species. Individuals of this species were bred in the laboratory of the Zoology Team at the University of Silesia in Katowice, Poland. A description of the morphology of the nymphs and eggs is presented. In addition to descriptions, photos of the successive immature stages are provided and scanning electron microscopy (SEM) images are included to show morphological details and compare the developmental changes in subsequent stages.

## 1. Introduction

When looking for comparative scientific information in journals, it can be concluded that, compared to the number of insect species described, the number of articles on nymphal stages is insufficient. Even within the suborder Heteroptera, the large group represented by the family Reduviidae Latreille, 1807 [1] still lacks information about the morphology of the eggs and nymphal stages. Species within subfamily Triatominae seem to be the most widely described in this respect, which is undoubtedly influenced by these insects’ role in spreading Chagas disease [2,3]. However, knowledge of the morphology of the nymphal stages and eggs has its application in the phylogenetic and systematic studies of a given group [4,5]. Therefore, increasing the scope of information in this area is advisable and desirable.

The Reduviidae currently consists of 24 subfamilies [2]. Among them, data on the morphology of nymphs and/or eggs has been published only for representatives of nine subfamilies, as follows. However, many are represented by only single reports: Ectrichodiinae Amyot and Serville, 1843 (eggs [6], *Neohaematorrhophus therasii* Ambrose and Livingstone, 1986 [7], *Labidocoris* Mayr, 1865 sp.—only eggs [8]), Emesinae Amyot and Serville, 1843 (*Liaghinella andina* Forero, 2007 [9], *Metapterus linearis* A. Costa, 1863 [10], *Pseudometapterus umbrosus* (Blatchley, 1926) [11]), Harpactorinae Amyot and Serville, 1843 (e.g., eggs [8,12], *Apiomerus crassipes* (Fabricius, 1803) [13], *Arilus gallus* (Stål, 1872) [14], *Endochus migratorius* Distant, 1903 [15], *E. umbrinus* Distant, 1902 [16], *Irantha armipes* (Stål, 1855) [17], *Manicocoris rufipes* (Fabricius, 1787) [18], *Pselliopus barberi* Davis, 1912 and *P. cinctus* (Fabricius, 1776) [19], *Polididus armatissimus* Stål, 1859 [20], *Rhaphidosoma atkinsoni* Bergroth, 1893—only eggs [21], *Repipta* Stål, 1859 spp. [22], *Rhynocoris kumarii* Ambrose and Livingstone, 1986 [23], *Sinea complexa* Caudell, 1900 [24,25], *S. diadema* (Fabricius, 1776) [25,26], *S. spinipes* (Herrich-Schaeffer, 1846) [25,27], *Sphedanolestes minusculus* Bergroth, 1908 [28], *Sycanus pyrrhomelas* Walker, 1873 [29], *Vachiria deserta* (Becker, 1867) [8], *Vesbius sanguinosus* Stål, 1874 [30], *Zelus leucogrammus* (Perty, 1833) [31], *Z. longipes* (Linnaeus, 1767) [32], *Z. socius* (Uhler, 1872) [33]), Peiratinae Amyot and Serville, 1843 (eggs [8,34], *Ectomocoris tibialis* Distant 1902 [35], *E. vishnu* Distant, 1904 [36], *E. xavierei* Vennison and Ambrose 1990 [37]), Reduviinae Latreille, 1807 (e.g., eggs [8], *Acanthaspis siva* Distant, 1904 [38], *Platymeris biguttatus* (Linnaeus, 1767) [39], *Reduvius personatus* (Linnaeus, 1758) [40]), Salyavatinae Amyot and Serville, 1843 (*Paralisarda malabarica* Miller, 1957 [41]), Stenopodinae Amyot and Serville, 1843 (*Oncocephalus annulipes*—only eggs [8], *O. plumicornis* [10]), Triatominae Jeannel, 1919 (e.g., eggs [6], *Alberprosenia goyovargasi* Martínez and Carcavallo, 1977 [42], *A. malheiroi* Serra, Atzingen and Serra, 1987 [43], *Belminus herreri* Lent and Wygodzinsky, 1979 [44], *Belminus* Stål, 1859 spp.—eggs [45], *Dipetalogaster maxima* (Uhler, 1894) [46], *Linshcosteus karupus* Galvão, Patterson, Rocha and Jurberg, 2002 [47], *Panstrongylus geniculatus* (Latreille, 1811) [48,49], *Rhodnius* Stål, 1859 spp. [50,51,52,53,54,55], *Triatoma* Laporte, 1833 spp. (e.g., [3,56,57,58,59,60,61,62,63,64,65,66])), eggs structure in general for several species of Reduviidae [67].

About 140 genera have been described within the subfamily Reduviinae. Most species are nocturnal, so their lifecycles are generally poorly known. They are considered generalist predators of insects and other arthropods [2,68]. The representatives of the genus *Psyttala* Stål, 1859 [69], have the largest body sizes in the subfamily Reduviidae. So far, six species have been described, all distributed exclusively in Africa: *P. ducalis* (Westwood, 1845) [70], *P. dudgeoni* Distant, 1919 [71], *P. horrida* (Stål, 1865) [72], *P. incognita* Distant, 1919 [71], *P. johnstoni* Distant, 1919 [71] and *P. samwelli* Distant, 1919 [71]. It should be noted here that the name *Psytalla* often used recently is incorrect, and the original spelling should be restored (see Results: Taxonomy).

*Psyttala horrida* (commonly called “the horrid king assassin bug” or “giant spiny assassin bug”) is a popular species in breeding, which can be seen on terroristic events and websites where insects are sold. As we have individuals of this species in our breeding, we decided to describe the nymphs and eggs. It will allow the completion of knowledge about the immature forms of Reduviidae, and the provided descriptions can be used in research on the phylogenetic position within the genus or even family.

## 2. Material and Methods

### Morphological Methods

The description of the eggs and all five immature stages was based on 30 specimens of *Psyttala horrida* bred in the Laboratory of Applied Entomology and Insectarium located at the Institute of Biology, Biotechnology, and Environmental Protection, Faculty of Natural Sciences, the University of Silesia in Katowice (Poland), as well as specimens housed in the Natural History Museum in Paris (MNHN). The breeding program was carried out in a glass terrarium, on a coconut substrate, in constant temperature conditions maintained at the level of 28–30 °C and humidity at the level of approximately 65–75%. Studies were conducted based on morphological and metric features.

A Nikon SMZ 25 stereoscopic microscope and a Nikon Eclipse NiU microscope were used to observe the external morphology. Measurements are given in centimetres and millimetres. Dead specimens were preserved in 70% alcohol, and a procedure described by Kanturski et al. [73] that was slightly modified was applied to dehydrate the material: serial baths of 80%, 90% and 96% ethanol for 40 min each (as the specimens were large). Dehydrated eggs and nymphs were dried in Leica EM CPD300 critical point dryer. Such prepared materials were placed on aluminium stubs with double-sided carbon tape (sometimes Leit-C plast conductive adhesive paste was also used) to stabilise. Then the material was sputtered with 30 nm gold using the turbomolecular pump coater (Quorum 150T ES plus—Quorum Technologies, Laughton, East Sussex, United Kingdom). After this, specimens were observed with the use of the scanning electron microscopes (Phenom XL Phenom-World BV the Netherlands and Hitachi UHR FE-SEM SU 8010 High Technologies, Tokyo, Japan) (low vacuum conditions at 10, 15, and 20 accelerating voltages with secondary electron detector) in the scanning microscopy laboratory of the Faculty of Natural Science, Institute of Biology, Biotechnology and Environmental Protection of Silesian University in Katowice. For structures observed with a scanning electron microscope (SEM), measurements are given in micrometres. Figures were prepared using NIS Elements D, Image Composite Editor software ver. 2.0.3.0 (panoramic image stitcher) (Microsoft Corporation, Redmond, WA, USA), PhotoScapeX ver. 4.2.1 (Mooii Tech, South Korea) and FireAlpaca 2.8.10 (PGN Inc., Japan).

The terminology for external morphology follows Schuh and Weirauch [2], and that for external structures of the dorsal abdominal glands follows Weirauch [74]. Eggs terminology follows Haridass [6]. Sensilla terminology and classifications follow Ahmad et al. [75].

## 3. Results

### 3.1. Taxonomy

Subfamily Reduviinae Latreille, 1807 [1]

Genus ***Psyttala*** Stål, 1859 [69]

***Psyttala*** Stål, 1859, 16:187 [69]. As subgenus of *Platymeris*. Type species: *Platymeris* (*Psyttala*) *ducalis* Westwood 1845, 4:120 [70]. By monotypy.

*Pysttala* (*sic*) Distant, 1919, 1:466 [71]

*Psytalla* (*sic*) Maldonado, 1990, 432 [76]

***Psyttala horrida*** (Stål, 1865), 3:123 [72]. Holotype (♂): Nigeria: Calabar; NHMW—Naturhistorisches Museum Wien, Wien, Austria [77].

The name of this genus was confused by Distant in 1919 [71], where he described it as *Pysttala*. Then a mistake repeated often by other scientists was made by Maldonado in 1990 [76], misspelling the name as *Psytalla*. All previous works describing this genus, including the original work where it was designated, use the name *Psyttala*. Therefore, restoring the proper name is appropriate and consistent with the International Code of Zoological Nomenclature (ICZN), especially on websites and forums where individuals of this genus are sold.

### 3.2. Morphological Research

#### 3.2.1. Eggs Structure

Females of *P. horrida* lay eggs singly. Eggs are symmetrical and ellipsoid, and brilliant dark brown (Figure 1). The operculum directly contacts the chorion, forming a groove between the chorial and opercular borders (Figure 2A). The exochorion has irregular ornamentation, sometimes comprised of quadrangular and pentagons, less prominent towards the posterior pole (Figure 2B). No holes have been identified on the chorion surface, neither micropyles nor aeropyles. The operculum is circular, slightly convex, and white; with polygonal ornamentation, mostly pentagons and hexagons were evident. The surface of the operculum is almost entirely covered with numerous orifices (Figure 2C). The mean total length from the opercular margin to the posterior pole is 3.05 ± 0.1 (3.45 mm with operculum). Maximum width at the broadest point is 2.0 ± 0.1. The diameter of the operculum is 1.46 ± 0.04.

#### 3.2.2. Description of Nymphs

##### First Instar

Colouration: Head and thorax dark brown to black, thorax with brighter ventral part (brown to red) and light pink ecdysial lines. Eyes silver. Scape and pedicel dark brown. Basiflagellomere with dark brown basal part and gradually paler into the apex, distiflagellomere greyish. All antennomeres covered with brown setae. Legs with red coxa and brownish-red trochanter. Femur yellow with brownish-red basal and red apical part, covered with light brown setae. Tibia yellow with red basal and pale brown apical part, covered with yellowish setae. Tarsus pale brown. Abdomen red to dark pink with black sclerotised muscle attachments and plates on the dorsal part. Tip of the abdomen dark brown to black (Figure 3).

Structure: Body dull with shiny head and thorax, covered by various-sized setae (Figure 4). Head with medium-sized and long setae. Ecdysial line visible in lateral parts of transversal suture of the head (Figure 3). Regarding the postocular area, the middle part of the ecdysial line is u-shaped and connected with a longitudinal ecdysial line placed in 1/2 of the postocular part and on the neck. Eyes prominent. Clypeus covered by long setae, mandibular and maxillary plates covered by medium-sized setae. Labrum visible along the whole length of the first visible labial segment. All labial segments covered by long setae (Figure 5A,B). Stridulitrum well developed, in the form of a gutter. Stridulatory furrow with regular stripes and flanked by lateral papillae (Figure 5B,D). Scape short and wide. Pedicel slightly thinner than the scape, both segments covered by long, erect, and robust setae. End of pedicel with single antennal *trichobothrium*. Basiflagellomere the longest, covered by long, erect, and regularly placed setae. Distiflagellomere, covered by few long, erect setae and short, dense, adherent setae (except 1/6 of basal part). Neck without setae. Pronotum covered by a few long, regularly placed setae. Mesonotum with two long setae (Figure 5C). Propleuron without setae, mesopleuron with two long setae on each side, and metapleuron with one seta on each side. Prosternum with two long setae near stridulitrum. Metanotum without setae in dorsal view. The whole thorax covered with short and robust spine-like processes (in this stage, visible at very high magnification (Figure 6A). Coxa with long, arranged in regular line setae (Figure 6B). Ventral apical part of fore- and mid-tibia covered by a pad of barbed setae (Figure 6C). Foretibial comb not on the spur (Figure 6C,D). On each abdominal segment, visible long setae placed on distinct sclerotised plates and arranged in transversal rows—four setae on each segment, except for the row on the first abdominal segment, which contains only two setae. A pair of long setae visible on lateral margins of each abdominal segment. In dorsal view, large evaporatory area of dorsal abdominal glands visible between segments III/IV, IV/V, and V/VI. Segments IX and X with visible sclerotised plates of tergites. Ventral side of the abdomen covered by transversal rows of long and thin setae (one row on each segment) without any sclerotised elements.

Measurements (in mm): Body length 7.6–8.1; maximum width of abdomen 3.1–3.27; head length 1.72–1.75; head width 0.97–1.04; length of anteocular portion 0.82–0.86; length of postocular portion 0.54–0.6; length of synthlipsis 0.73–0.76; length of antennal segments I, II, III, IV, are 0.78–0.86, 1.77–1.81, 3.16–3.37, and 2.89–2.99, respectively; length of labial segments I, II, and III are 0.8, 0.7, and 0.5, respectively; maximum length of pronotum 0.89–0.95; maximum length of mesonotum 0.87–0.97; maximum width of pronotum 1.4–1.48; maximum width of mesonotum 1.57–1.62; maximum of abdomen length 3.89–4.4.

##### Second Instar

Colouration: Head and thorax same as in the first instar nymphs. Antennomeres with very dark brown scape and pedicel. Basiflagellomere with dark brown basal part gradually paler into the apex, distiflagellomere brown. All segments covered with brown setae. Legs with brownish-red coxa and brown trochanter. Femur yellow with dark brownish-red basal and red apical parts, covered with light brown setae. Tibia yellow with red basal and pale brown apical parts, covered with yellowish setae. Tarsus pale brown. Abdomen brownish-dark pink with red and dark brown muscle attachments, plates, and spine-like processes on the dorsal part. Last segment of the abdomen dark brown. Single red spot in the central part of the fourth and fifth abdominal segments, seen ventrally (Figure 7).

Structure: Anteocular part of the head with four large processes placed between the eyes in dorsal view (interior pair visibly larger); postocular with two medium-sized processes. Prosternum with two large and robust processes, with globularly rounded apices in the anterior part, placed laterally to stridulitrum. Robust processes visible on pronotum: two processes placed laterally on collar; I transversal row: two large processes placed near longitudinal suture; II transversal row: four large processes (II inner larger than two outers (placed on lateral margin)); III transversal row: four large processes (inner ones smaller). Mesonotum: I transversal row: two lateral, medium-sized processes; II transversal row: four small processes; III transversal row: four large processes (inner ones larger). Mesonotum with two large lateral processes; visible crescent, sclerotised plate of the metathorax. Pad of barbed setae covering 1/3 of the apical part of fore- and midtibiae (delicately larger on foretibiae). Legs covered by relatively long and dense setae. First abdominal segment in dorsal view with visible spiracle and two small processes placed in the middle; II–VIII with visible sclerotised parts: plate with a process on the lateral edge, a process in the middle part, and large process near the longitudinal axis of the abdomen. Genital segments sclerotised. Evaporatory area of dorsal abdominal glands placed on globular enlargements, distinctly visible. Ventral surface of abdomen with visible, sclerotised muscle attachments. Middle part of IV–V abdominal segments with small sclerotisations. Abdominal segments VI–VII ventrally with large sclerotisations (Figure 7 and Figure 8).

Measurements (in mm): Body length 10.8–11.5; maximum width of abdomen 4.64–4.75; head length 2.26–2.38; head width 1.16–1.21; length of anteocular portion 1.11–1.14; length of postocular portion 0.72–0.75; length of synthlipsis 0.92–0.95; length of antennal segments I, II, III, and IV are 0.98–1.1, 2.5–2.7, 4.13–4.17, and 3.69–3.78, respectively; length of labial segments I, II, and III are 1.05, 0.89, and 0.56, respectively; maximum length of pronotum 1.32–1.38; maximum length of mesonotum 1.42–1.46; maximum width of pronotum 1.91–1.94; maximum width of mesonotum 2.4–2.45; maximum of abdomen length 5.9–6.1.

##### Third Instar

Colouration: Head and thorax black in dorsal view. Scape and pedicel dark brown, pedicel with black apical apex. Basi- and distiflagellomere brownish with darker irregular spots. All antennomeres covered with brown and light setae. Legs yellowish; basal part of femur, coxa, and trochanter brown. Longitudinal ridge of meso- and metasterna as well as posterior ridge of mesosterna whitish. Abdomen and ventral parts of the head and thorax brown to dark brown. Abdomen ventrally pinkish (Figure 9).

Structure: Whole head more sclerotised, ecdysial lines thinner. Ante- and postocular part of the head with same number and placement of spine-like processes, larger than in second instar. Thorax dorsally covered by small granulations on most of its surface. Compared to the previous stage, the pronotum with visibly elongated large and robust processes, with a delicately elongated apical part. Spine-like processes visible on the pronotum with the same number and placement as in second instar but visibly larger. Spine-like processes mesonotum: I transversal row: two lateral, medium-sized processes; II transversal row: four processes—two long, lateral, placed on the edge or lobe, and two very small, visible in the middle; III transversal row: four large processes (inner ones larger) with additionally two small lateral processes visible on the edge of the lobe. Lateral plates of mesonotum with two large lateral processes and two medial, distinctly visible granules. Pad of barbed setae covering 1/3 of the apical part of foretibia and 1/4 of midtibia. Legs covered by relatively long and dense setae, distinctly denser on the ventral part of fore- and midfemur and tibia. The number and the placement of spine-like processes on the abdomen are the same as in second instar, except they are visibly longer. Three evaporatory areas of dorsal abdominal glands placed on globular enlargements, distinctly visible but not so sclerotised as other elements of tergites. Ventral surface of abdomen with visible, sclerotised muscle attachments. Middle part of IV–V abdominal segments with small sclerotisations. Abdominal segments VI–VII with large sclerotised areas ventrally (Figure 9 and Figure 10).

Measurements (in mm): Body length 15.8–17.7; maximum width of abdomen 6.8–7.4; head length 3.45–4.0; head width 1.7–2.1; length of anteocular portion 1.7–1.85; length of postocular portion 1.19–1.55; length of synthlipsis 1.3–1.49; length of antennal segments I, II, III, and IV are 1.45–1.85, 3.75–4.7, 5.1–5.3, and 3.9–4.1, respectively; length of labial segments I, II, and III are 1.75, 1.62, and 1.05, respectively; maximum length of pronotum 2.02–2.64; maximum length of mesonotum 2.88–3.8; maximum width of pronotum 2.23–2.9; maximum width of mesonotum 3.95–5.17; maximum of abdomen length 8.9–9.7.

##### Fourth Instar

Colouration: Head and thorax black, except pale ventral side of mesothorax (mesotergite with pale longitudinal ridge and pale posterior edge). Scape and pedicel black. Basi- and distiflagellomere dark brown. All antennomeres covered with brown and light setae. Legs yellowish; basal part of the femur, coxa, and trochanter brown. Abdomen and ventral parts of the head and thorax brown to dark brown. Abdomen brown, ventrally pinkish; laterotergites of abdominal segment II black; laterotergites of segments V–VII with yellow anterior and black posterior half. All abdominal spine-like processes black (Figure 11).

Structure: Whole head strongly sclerotised with distinct granulation and visible, thin ecdysial lines. Ante- and postocular part of the head with same number and placement of spine-like processes, larger than in third instar. Thorax dorsally covered by small granulations on all its surface. Compared to the previous stage, the prosternum with visibly elongated large and robust processes, with delicately elongated apical part. Spine-like processes visible on the pronotum with same number and placement as in third instar, but visibly larger; however, the middle processes of last row strongly elongated compared to third instar. Spine-like processes on mesonotum: I transversal row: two lateral, medium-sized processes; II transversal row: four processes—two long, lateral, placed on the edge or lobe, and two small (longer than in third instar), visible in the middle; III transversal row: four large processes (inner ones visibly larger, and some specimens with different length of those) and additionally, two medium-sized lateral processes visible on the edge of the mesonotum. Lateral plates of mesonotum with two long lateral processes and two with distinct granulations on the surface. Sclerotised, lateral plates of metathorax with one long spine-like process. Pad of barbed setae covering 1/3 of the apical part of foretibia and 1/4 of midtibia. Legs covered by relatively long and dense setae, distinctly denser on the ventral part of fore- and midfemur and tibia. Fore- and midlegs (femur and tibia) with single trichobothrial setae. Number and placement of the spine-like processes on the abdomen are same as in third instar, except they are visibly longer. Laterotergites of abdominal segments II–VII with three visible spine-like processes on the outer edge, variable in length. On laterotergites II–VI, the posterior process the longest; laterotergite VII with short processes, similar in length. Three evaporatory areas of dorsal abdominal glands visible between segments III–VI. Spiracle on first segment very small, evaporatory areas placed on globular enlargements, distinctly visible. Ventral surface of abdomen with visible, sclerotised muscle attachments. Middle part of IV–V abdominal segments with small sclerotisations. Abdominal segments VI–VII with large sclerotised areas ventrally (Figure 11 and Figure 12).

Measurements (in mm): Body length 22.4–23.2; maximum width of abdomen 9.2–9.7; head length 4.2–4.8; head width 2.2–2.38; length of anteocular portion 1.88–2.3; length of postocular portion 1.3–1.8; length of synthlipsis 1.5–1.65; length of antennal segments I, II, III, and IV are 2.3–2.6, 4.5–7.0, 5.55–6.7, and 4.1–4.95, respectively; length of labial segments I, II, and III are 2.0, 1.7, and 1.25, respectively; maximum length of pronotum 2.8–3.85; maximum length of mesonotum 3.5–4.6; maximum width of pronotum 4.3–5.5; maximum width of mesonotum 5.8–6.9; maximum of abdomen length 10.6–12.2.

##### Fifth Instar

Colouration: Head and thorax black, except pale middle part of tergite of metathorax, ventral side of mesothorax (mesotergite with pale longitudinal ridge). Scape and pedicel black. Basi- and distiflagellomere dark brown. All antennomeres covered with brown and light setae. Forefemur with red basal part, with two dark spots on the dorsal surface and dark stripe on the ventral surface (reaching half its length). Midfemur yellow with 1/3 of the basal part black. Hind femur yellow with black basal and apical parts. Foretibia yellow. Midtibia yellow with darker apical part. Hind tibia yellow with 1/3 of the basal and apical part dark. Abdomen dorsally brown, ventrally yellowish/greyish; laterotergites of abdominal segment II black; laterotergites of segments V–VII with yellow anterior and black posterior half. All abdominal spine-like processes black. Abdominal muscle attachments black (Figure 13).

Structure: Whole head strongly sclerotised with distinct granulation and visible; ecdysial lines black, hardly visible. Ante- and postocular part of the head with same number and placement of spine-like processes with similar length than in fourth instar. Thorax dorsally covered by small granulations on all its surface. Compared to the previous stage, the prosternum with visibly elongated large and robust processes, with a delicately elongated apical part. Spine-like processes visible on the pronotum with the same number and placement as in fourth instar but visibly larger; however, the collar processes are shorter and more robust than in fourth instar, and the lateral processes of posterior row in some specimens are with divided apices. Spine-like processes on the mesonotum are rearranged and change their placement: two long processes are placed in the middle of the mesonotum, and anteriorly of them, two very small granules visible. On the coastal ridge of wing pads, there are visible 6–7 (variable number) processes, 3–4 (I, II, IV and VI) are distinctly longer than others, and the first one (in the basal part of the wing pad) in some cases with divided apex. Lack of spine-like processes of metanotum. Pad of barbed setae covering 1/3 of the apical part of foretibia and 1/4 of midtibia. Legs covered by relatively long and dense setae, distinctly denser on the ventral part of fore- and midfemur and tibia. Fore- and midlegs (femur and tibia) visible a single trichobothrial setae. Number and placement of the spine-like processes on the abdomen are same as in fourth instar. Laterotergites of abdominal segments II–V with one large visible spine-like process visible on the outer edge, variable in length. On laterotergites II–VI, posterior process the longest; laterotergite VII with short processes, similar in length. Three evaporatory areas of dorsal abdominal glands visible between segments III–VI. Area between segments II/III is very small (smaller than in fourth instar), evaporatory area between segments III–VI (V–VI the smallest) distinctly visible on globular enlargements. Ventral surface of abdomen with visible, sclerotised muscle attachments. Middle part of IV–V abdominal segments with small sclerotisations. Abdominal segments VI–VII with large sclerotised areas ventrally (Figure 13 and Figure 14).

Measurements (in mm): Body length 28.1–30.55; maximum width of abdomen 10.7–11.25; head length 5.05–5.46; head width 2.7–3.0; length of anteocular portion 2.5–2.9; length of postocular portion 1.8–2.05; length of synthlipsis 1.75–1.85; length of antennal segments I, II, III, and IV are 2.4–2.8, 7.0–7.25, 6.7–6.9, and 4.8–5.1, respectively; length of labial segments I, II, and III are 2.25, 1.8, and 1.35, respectively; maximum length of pronotum 3.9–4.1; maximum length of mesonotum 4.4–5.45; maximum width of pronotum 5.7–5.9; maximum width of mesonotum 7.1–7.5; maximum of abdomen length 16.1–17.0.

#### 3.2.3. Comparison of Morphological Structures

The morphological analysis of the successive immature stages of *P. horrida* shows that the growth of the body is proportional. Long appendages that cover the body are one of the most distinctive features of this species. The processes on the dorsal side of the head, thorax, and abdomen appear as long setae only in the first nymph (Figure 15A,D,G). Subsequently, they take the form of spine-like processes that grow larger with each stage (Figure 15B,C,E,F,H,I). Spine-like processes are often accompanied by a single (sometimes two), full-length cirrous seta placed on a distinct elevation with a strong cuticle sculpture (Figure 16A–C). Those processes are ribbed longitudinally at the top (Figure 16D). During moult, sometimes the appendages are not correctly developed; this causes their deformation, and sometimes even complete failure to develop (Figure 16E). On the edge of the abdomen, on each segment of the first nymph, there is a single seta (Figure 16F), and in the subsequent stages, an area is formed consisting of three appendages—one large and two smaller ending with setae (Figure 16G). After removing the apical part of the process, a hollow with a pore is visible (Figure 16H).

Except for the first nymph, at each following stage, four setae can be seen on the outer side of the antennae bases (Figure 17A). Small appendages are scattered densely on the dorsal side of the head, thorax, and abdomen (except the abdomen of the first nymph), ending with a hole at the top (Figure 17B,C). Successive segments of the antennae increase their length steadily in the following stages. The scape is the shortest and covered with long, erect, and robust setae with medium density. The pedicel is always slightly thinner but more than twice longer as the scape, also covered by long, erect, and robust setae. At the end of the pedicel, in each immature stage, a single antennal *trichobothrium* is visible (Figure 18B,C). The basiflagellomere is covered by very long, erect, and regularly but not densely placed setae (Figure 18A). The distiflagellomere at the basal part has only a few long, erect setae, but the rest of this segment is covered with short, dense, adherent setae and a few *sensilla coeloconica* between them (Figure 18D–F).

The stridulitrum is well developed from the first nymphal stage. In the following stages, it lengthens and slightly narrows (Figure 19A). The stridulatory furrow has regularly spaced grooves along the entire length (Figure 19C). Visible labial segments are covered with elongated *sensilla trichodea* (Figure 19B and Figure 20). On the fourth labial segment, *sensilla chaetica* and *trichoidea* can be identified (Figure 21A), and at the tip of the labium, two sensory fields are visible (Figure 21B). Right under the sensory field, a few *sensilla coeloconica* are located (Figure 21C,D). The apex of maxillary stylets is acute, while the subapical portion of maxillary stylets bears no blunt process but is covered with tiny ridges. The central part of maxillary stylets is bulging, with well-marked longitudinal grooves (Figure 22A,B). The mandibular stylets’ apex is covered with separate ridges and hooklets (Figure 22C).

From the first immature stage, three well-developed large evaporatory areas of dorsal abdominal glands are visible on the abdomen. In the first nymphal stage, evaporatory areas are located on the lower part of the III-IV-V abdominal segments, but in the following stages, those areas “go lower” and are located more between the III–IV, IV–V and V–VI segments. The last of them (on the border of V–VI segments) in each subsequent stage is smaller than the other two (Figure 4, Figure 8, Figure 10, Figure 12 and Figure 14). Spiracles on the dorsal side of the first abdominal segment are also modified during development. In the first and second immature stages, spiracles are round, then become elongated and grow larger with each stage (Figure 23A–D).

The immature stages of *P. horrida*, similarly to other representatives of the subfamily Reduviinae, possess a pad of barbed setae. In the case of this species, they are placed on the ventral apical part of fore- and midtibia. With each successive stage, the pad grows larger, occupying an average of 1/3 of the tibia (Figure 24A–C). It is composed of long, rigid setae with characteristic ribbing on one side (Figure 24D). Another interesting structure observed through scanning microscopy is an appendage near the eye on its ventral side (Figure 25A). The appendage appears with the second nymphal stage, and with each successive one, it becomes bigger. Under the forecoxa, some pointed appendages of unknown functions were identified (Figure 25B). At the lateral edge of the pronotum, a large spiracle-like structure on a tube-shaped base is visible (but only in the first and second instars; later, the construction of the pronotum makes it difficult to observe) (Figure 25C). In the fourth and fifth nymphal stages, dense, long setae can be observed on the middle trochanter, arranged in the form of two side strips (Figure 25D). Starting from the third instar, one small appendage forms on the distal part at the middle femur (Figure 25E). Whereas from the fourth instar, two small appendages form in the middle of the hind femur (Figure 25F).

## 4. Discussion

The availability of nymphal stages of various representatives of Reduviidae and the description of their development allows for a better understanding of this group. Using newer microscopic techniques, such as scanning electron microscopy, helps identify even the smallest changes in the development of nymphs [3,61,62,64,65]. 

The eggs of *Psyttala* and other Reduviinae [8,38,39,40] are most similar in shape to those found in Triatominae, especially *Alberprosenia goyovargasi*, *Linshcosteus karupus*, and *Triatoma* spp. [6,42,47,60,61,62,64,66]. The shapes of the eggs in the remaining subfamilies studied so far are different, but the greatest differentiation can be observed in the structure of the operculum. Within Reduviinae and Triatominae, the operculum is usually slightly convex, without any appendages, projections, depressions etc. Such variations are noticeable, for example, in Emesinae, Harpactorinae or Peiratinae [12,21,34,67]. In addition, we have not identified micropyles and aeropyles on the chorion. However, Cobben [67] suggested that the absence of micropyles could be explained by the fact that the eggs were already fertilised before the chorion was deposited. We do not know if this is the correct explanation, but we did not identify micropyles on the eggs of the two species of *Platymeris* and their hybrids either [78]. This issue in Reduviinae requires a better understanding.

A single *trichobothrium* at the end part of the pedicel is also shown in nymphs of other species of Reduviidae [61]. The sensillum looks similar in all studied species and has developed in the first nymphal stage. Furthermore, the stridulitrum develops in most species in all nymphal stages [62,65]; however, it is absent in the first nymphal stage of *Linshcosteus karupus* [47].

Small appendages which cover the dorsal side of the head, thorax, and abdomen (Figure 17B,C) look similar to the spine-like short-projection trichomes described by Weirauch [79]. This article found trichomes among the nymphal stages of species previously reported to possess camouflaging behaviour. However, we have not examined whether these structures secrete a sticky substance and can participate in camouflage. Moreover, under laboratory conditions, we did not observe nymphs using the substrate to hide.

The most characteristic feature of *P. horrida* nymphs and adults are appendages on the head, thorax, and abdomen. On the latter, they disappear in the adult form (Figure 15). The first nymphal stage has long setae in places, where the other stages have appendages. However, this seems logical, as rigid, and long appendages could interfere with embryonic development. We did not observe similar appendages in other nymphs of Reduviidae, except some representatives of Harpactorinae: *Pselliopus cinctus*, *P. barberi* [20], *Sinea complexa, S. diadema*, and *S. spinipes* [24,25,26,27]; however, the patterns of distribution and size of the appendages vary.

One of the features we have observed in the nymphs of *P. horrida*, which we have not identified in any other reviewed publication, is the spiracle-like structure on the lateral edge of the propleuron. However, it is possible that this structure is not rare but is rarely found. Locating it requires observation from the underside. Some specimens, especially those obtained from museum collections, are often attached to cardboard, making it impossible to analyse them as provided. Nevertheless, it is a structure that requires future analysis in other species.

## Figures and Tables

**Figure 1 insects-13-01014-f001:**
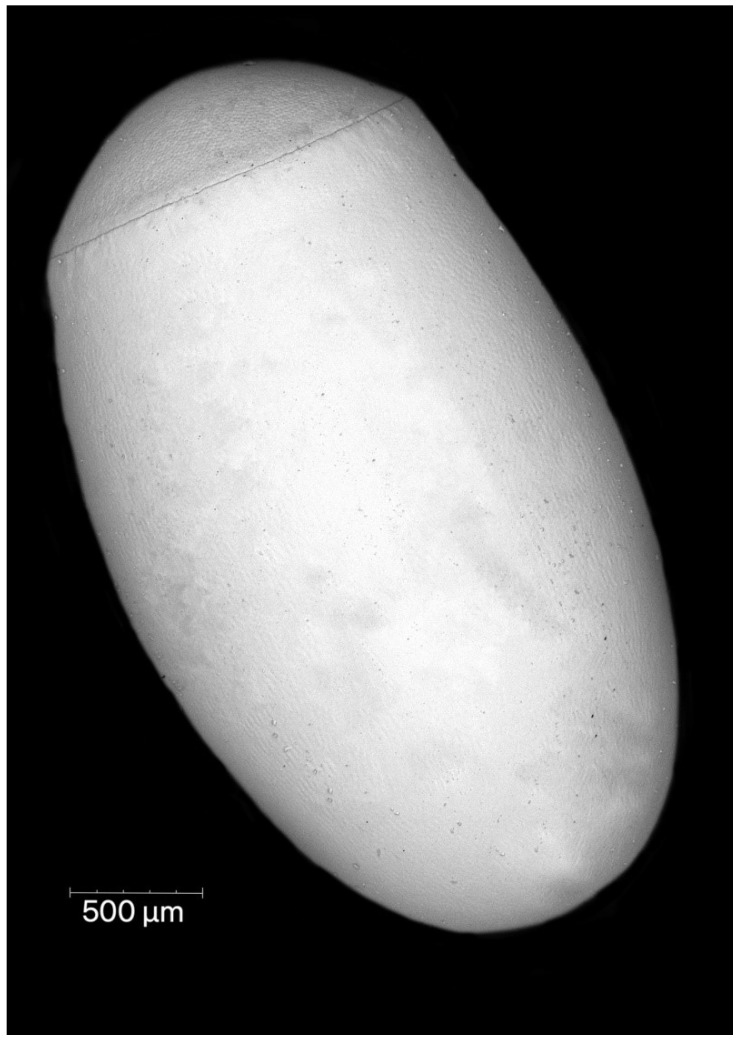
General view of the eggshell of *Psyttala horrida* (Stål, 1865).

**Figure 2 insects-13-01014-f002:**
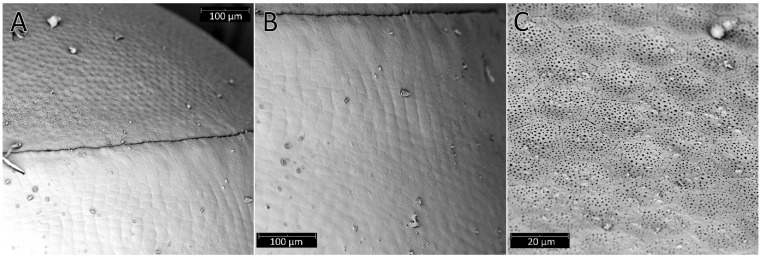
(**A**) The groove between the chorial and opercular borders *Psyttala horrida* (Stål, 1865); (**B**) The exochorion with irregular follicular imprints near the opercular region; (**C**) The numerous orifices in the polygonal ornamentation of the operculum.

**Figure 3 insects-13-01014-f003:**
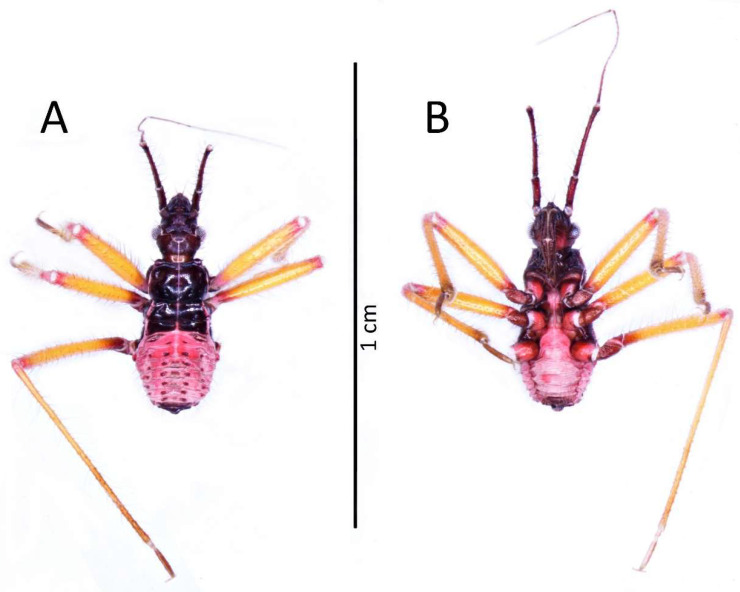
(**A**) Dorsal and (**B**) ventral habitus of the first instar of *Psyttala horrida* (Stål, 1865).

**Figure 4 insects-13-01014-f004:**
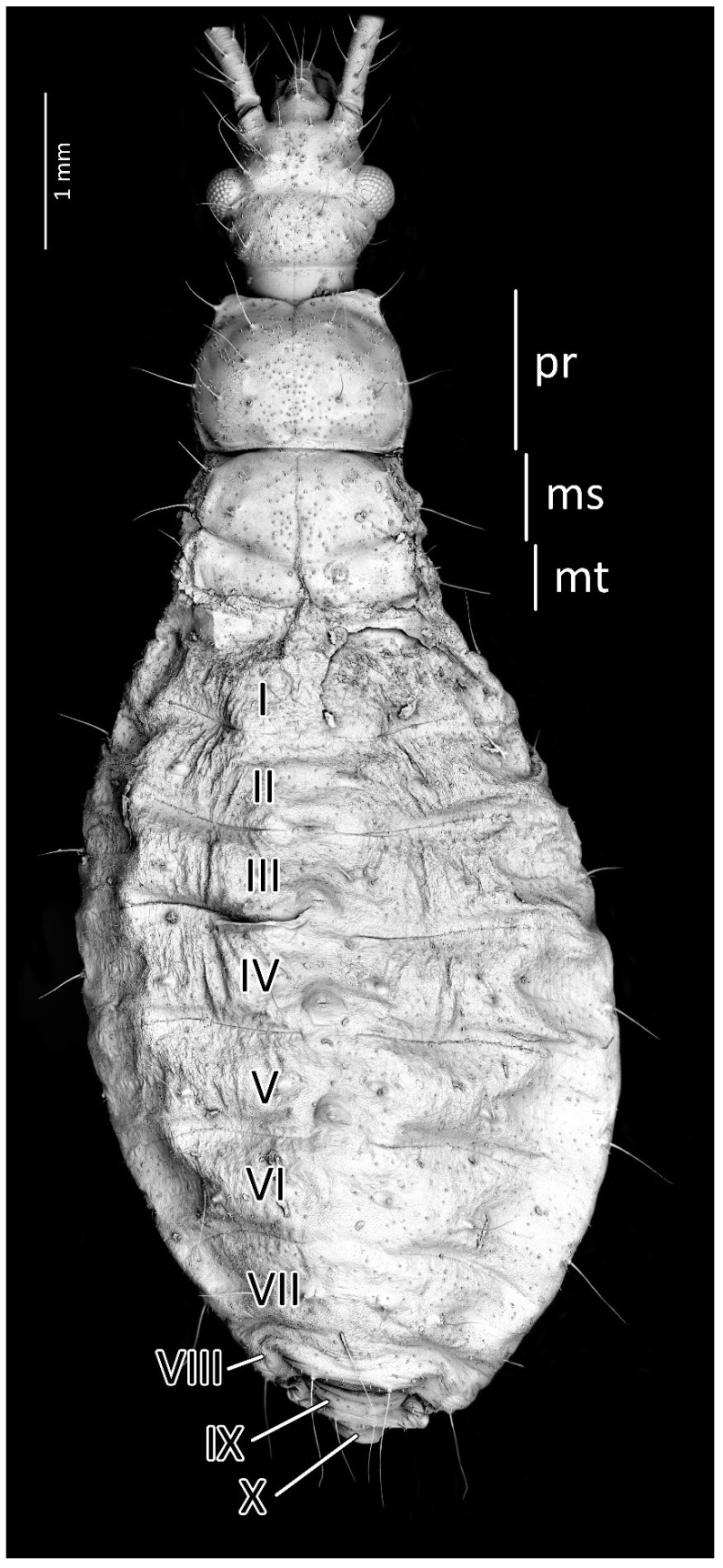
Dorsal habitus of the first instar of *Psyttala horrida* (Stål, 1865), detailed SEM view. **ms**—mesonotum; **mt**—metanotum; **pr**—pronotum; **I**–**X**—abdominal segments.

**Figure 5 insects-13-01014-f005:**
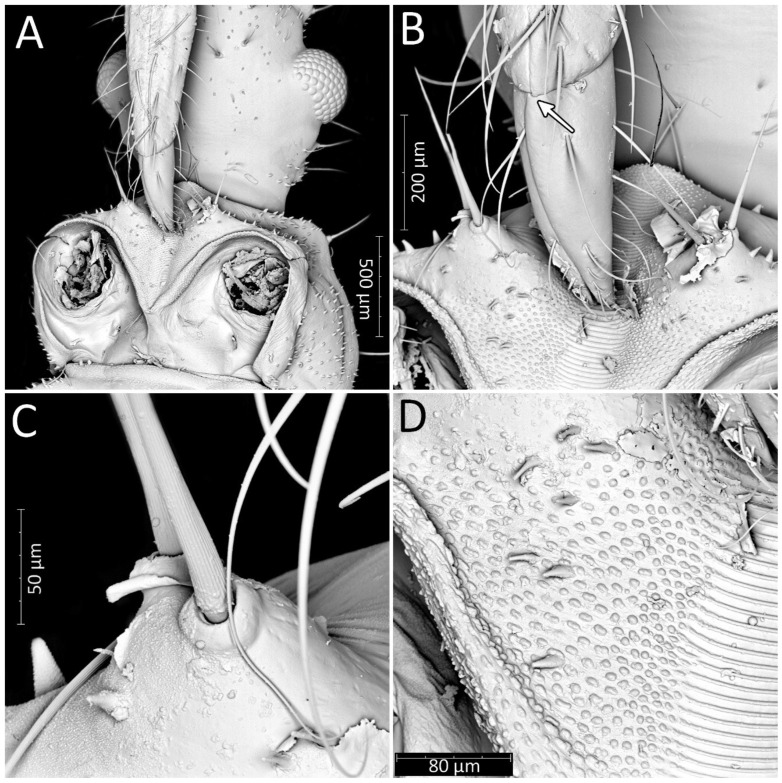
*Psyttala horrida* (Stål, 1865), first instar, SEM views. (**A**) Ventral view of the head with prosternum; (**B**) magnification of the tip of the labium and stridulitrum, with detail of membranous articulation between segments II and III; (**C**) magnification of two long setae near stridulitrum, with a clearly crimped structure; (**D**) stridulatory furrow with regular stripes and flanked by lateral papillae.

**Figure 6 insects-13-01014-f006:**
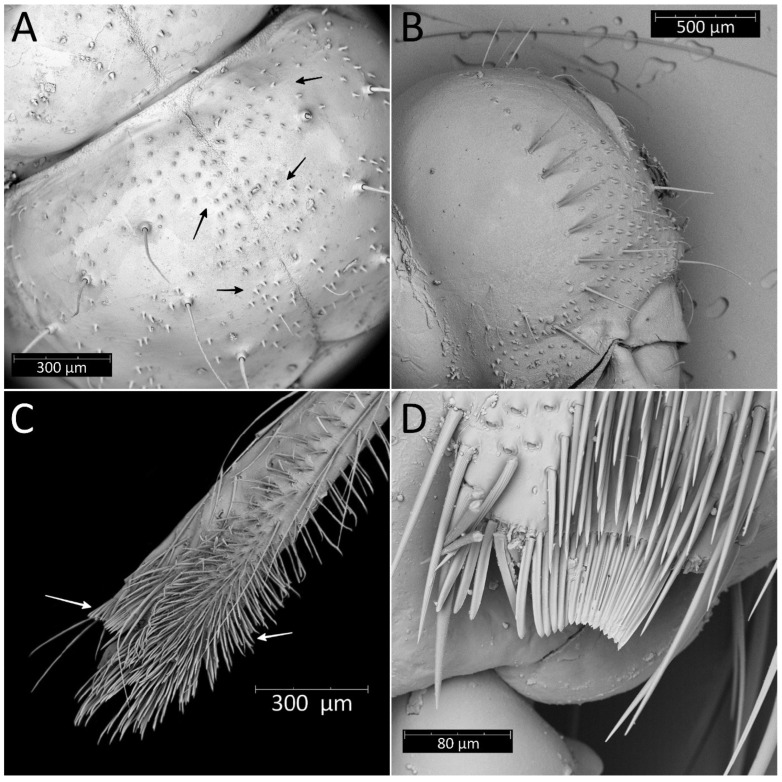
(**A**) Thorax of the first instar of *Psyttala horrida* (Stål, 1865) covered with short and robust spine-like processes; (**B**) coxa with long setae, arranged in the regular line, dorsal view; (**C**) pad of barbed setae on ventral apical part of fore- and middle tibia (arrow on the right); foretibial comb—arrow on the left; (**D**) foretibial comb, at larger magnification.

**Figure 7 insects-13-01014-f007:**
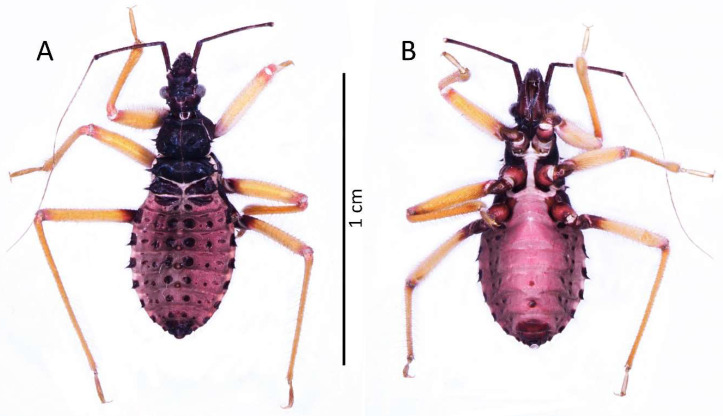
(**A**) Dorsal and (**B**) ventral habitus of the second instar of *Psyttala horrida* (Stål, 1865).

**Figure 8 insects-13-01014-f008:**
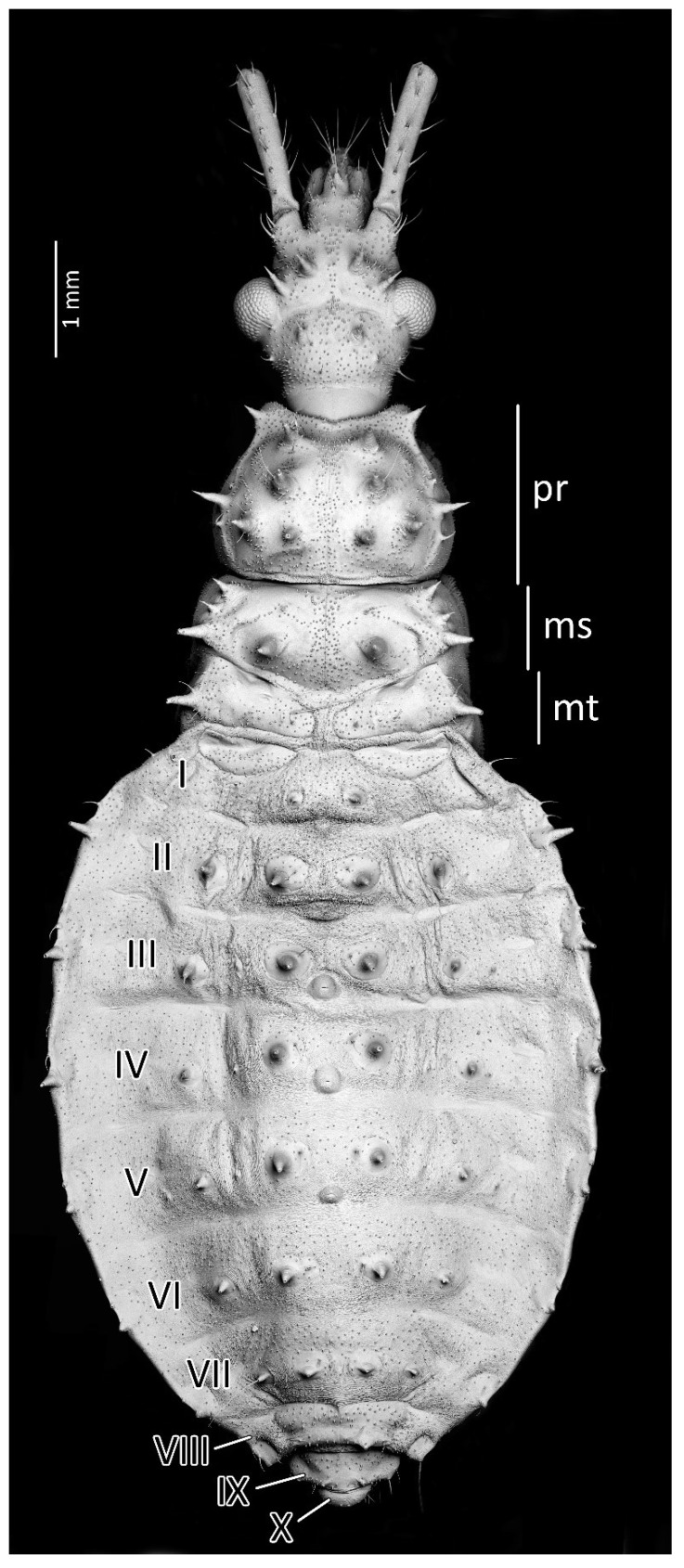
Dorsal habitus of the second instar of *Psyttala horrida* (Stål, 1865), detailed SEM view. **ms**—mesonotum; **mt**—metanotum; **pr**—pronotum; **I**–**X**—abdominal segments.

**Figure 9 insects-13-01014-f009:**
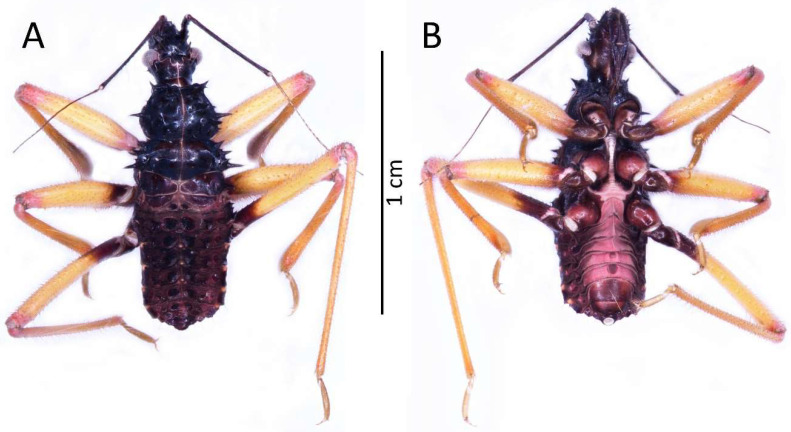
(**A**) Dorsal and (**B**) ventral habitus of the third instar of *Psyttala horrida* (Stål, 1865).

**Figure 10 insects-13-01014-f010:**
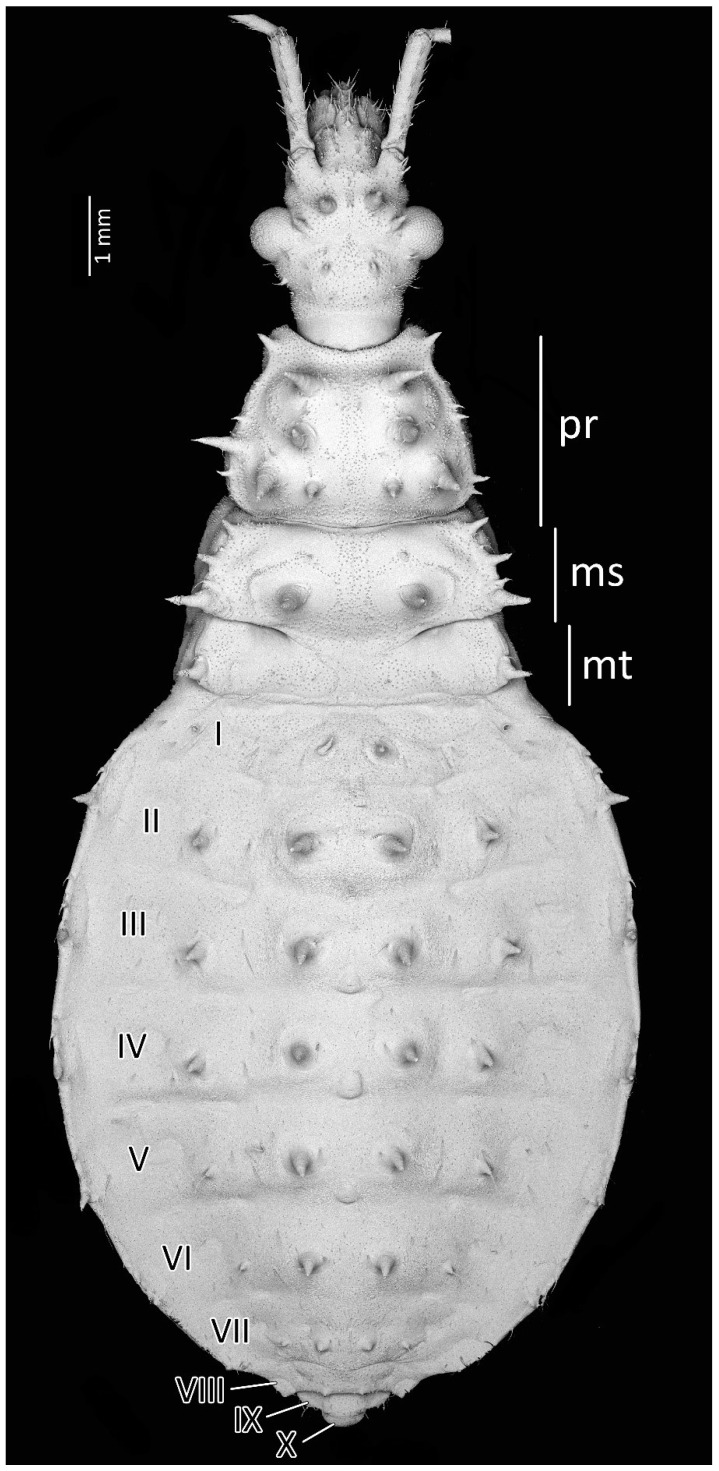
Dorsal habitus of the third instar of *Psyttala horrida* (Stål, 1865), detailed SEM view. **ms**—mesonotum; **mt**—metanotum; **pr**—pronotum; **I**–**X**—abdominal segments.

**Figure 11 insects-13-01014-f011:**
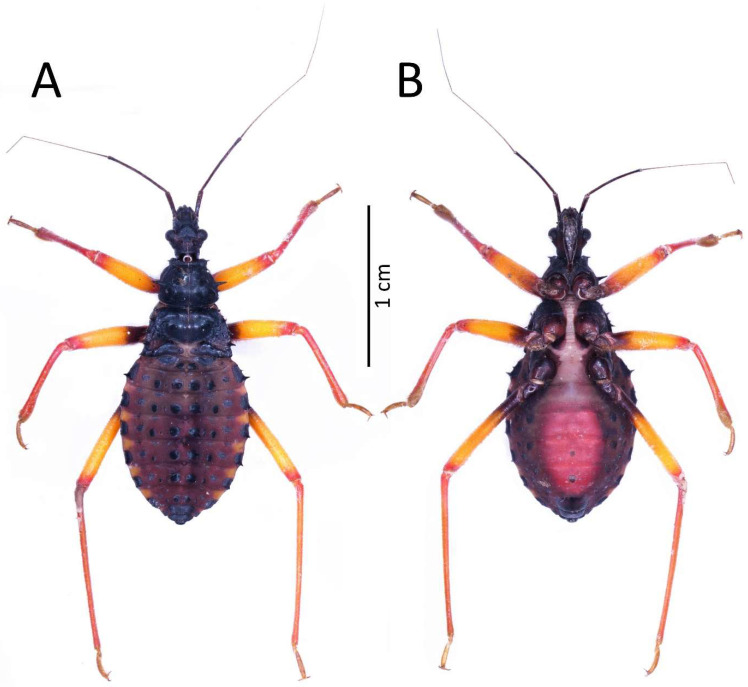
(**A**) Dorsal and (**B**) ventral habitus of the fourth instar of *Psyttala horrida* (Stål, 1865).

**Figure 12 insects-13-01014-f012:**
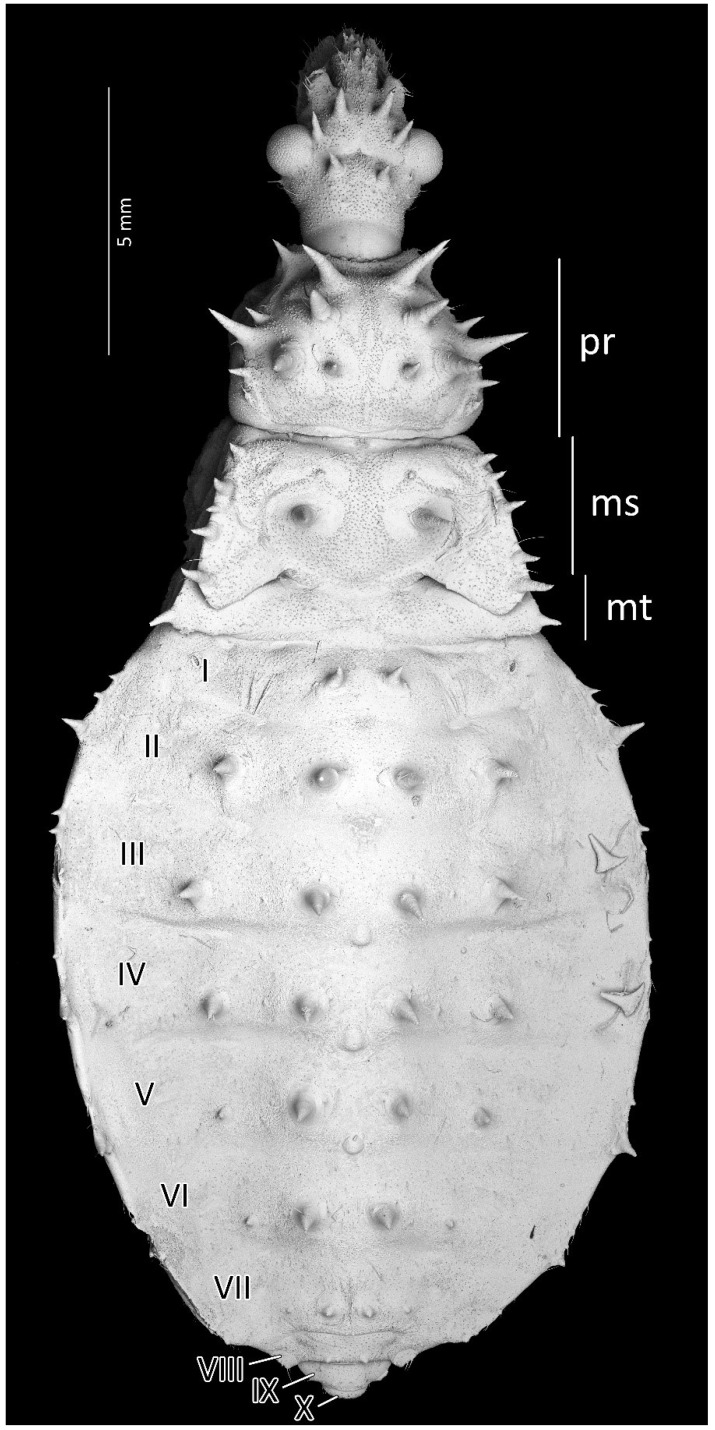
Dorsal habitus of the fourth instar of *Psyttala horrida* (Stål, 1865), detailed SEM view. **ms**—mesonotum; **mt**—metanotum; **pr**—pronotum; **I**–**X**—abdominal segments.

**Figure 13 insects-13-01014-f013:**
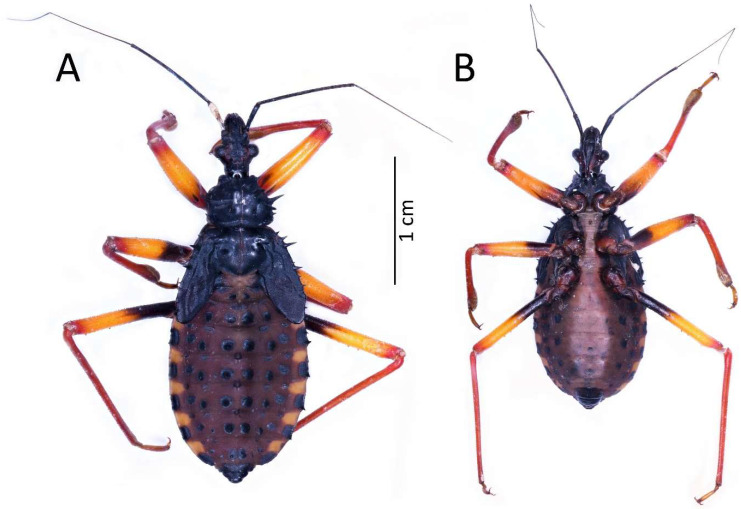
(**A**) Dorsal and (**B**) ventral habitus of the fifth instar of *Psyttala horrida* (Stål, 1865).

**Figure 14 insects-13-01014-f014:**
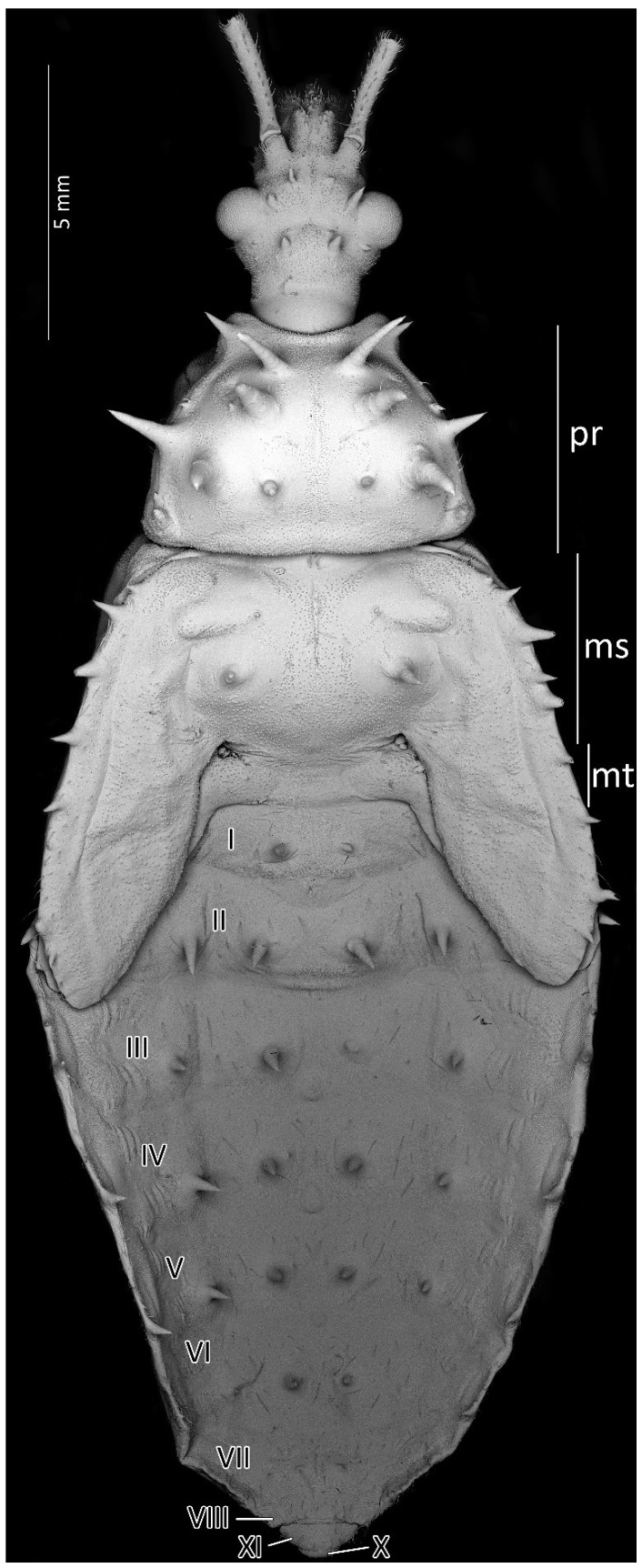
Dorsal habitus of the fifth instar of *Psyttala horrida* (Stål, 1865), detailed SEM view. **ms**—mesonotum; **mt**—metanotum; **pr**—pronotum; **I**–**X**—abdominal segments.

**Figure 15 insects-13-01014-f015:**
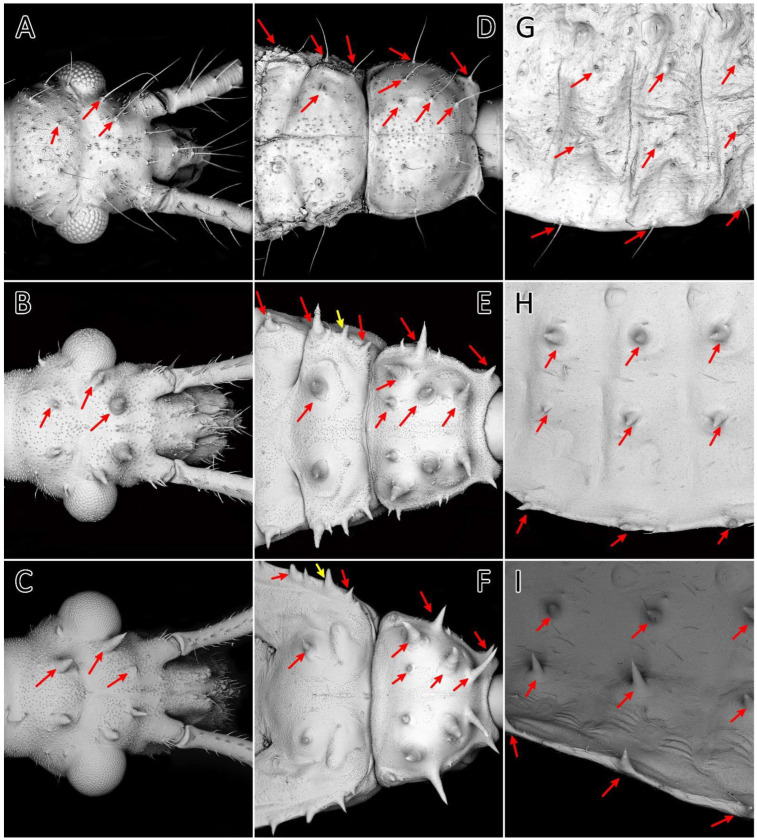
First (**A**,**D**,**G**), third (**B**,**E**,**H**), and fifth (**C**,**F**,**I**) instars of *Psyttala horrida* (Stål, 1865), SEM views. Long setae and corresponding spine-like processes on the dorsal side of the (**A**–**C**) head, (**D**–**F**) thorax, and (**G**–**I**) abdomen. The arrows indicate the corresponding appendages at the following stages.

**Figure 16 insects-13-01014-f016:**
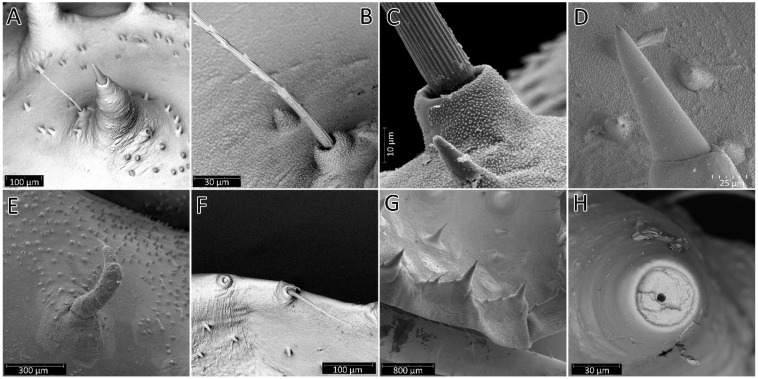
SEM views of the processes on the body of the instars of *Psyttala horrida* (Stål, 1865). (**A**) Spine-like processes accompanied by cirrous seta; (**B**) magnification of the cirrous seta; (**C**) magnification of the cirrous seta base with visible cuticle sculpture; (**D**) magnification of the longitudinally ribbed top of the process; (**E**) not correctly developed process; (**F**) first instar, single seta on the edge of the abdomen segment; (**G**) third instar, a field with three processes on the edge of the abdomen segment; (**H**) apical part of the process without a tip—visible pore in a hollow.

**Figure 17 insects-13-01014-f017:**
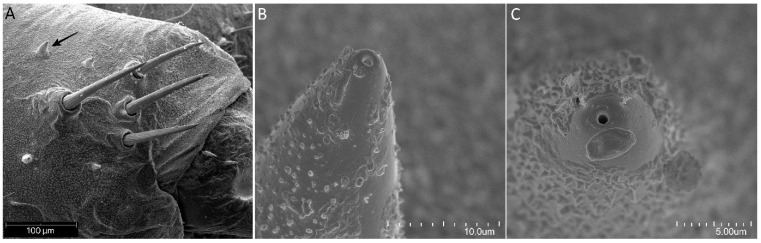
*Psyttala horrida* (Stål, 1865). (**A**) The antennal base with four trichoidea setae and small appendages [arrow]. (**B**) Magnification of the small appendage on the dorsal side of the head, thorax, and abdomen, lateral view and (**C**) dorsal view at the top with an apical pore.

**Figure 18 insects-13-01014-f018:**
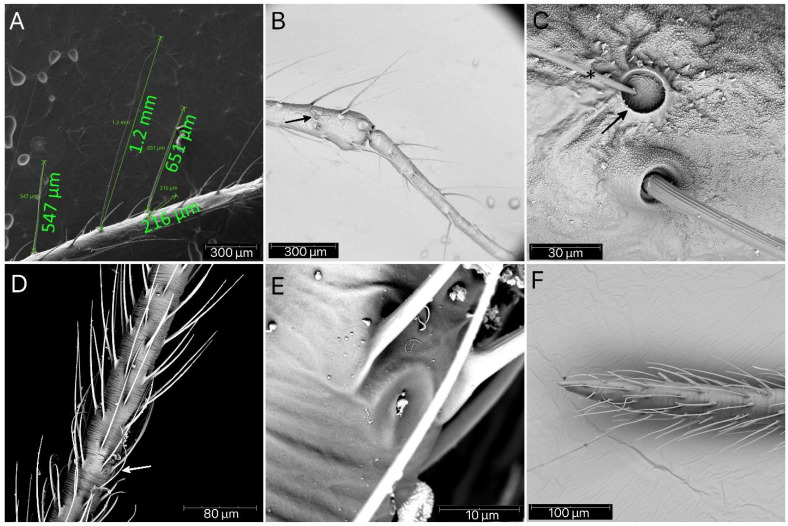
*Psyttala horrida* (Stål, 1865), antennae. (**A**) Setae of different lengths on the basiflagellomere; (**B**) the end part of the pedicel with a single *trichobothrium*, and (**C**) magnification of the antennal *trichobothrium* (mechanoreceptive seta [*****] situated in a depression in the cuticle [arrow]); (**D**) the central part of the distiflagellomere with densely arranged setae and (**E**) single *sensillum coeloconicum*; (**F**) apical part of the distiflagellomere.

**Figure 19 insects-13-01014-f019:**
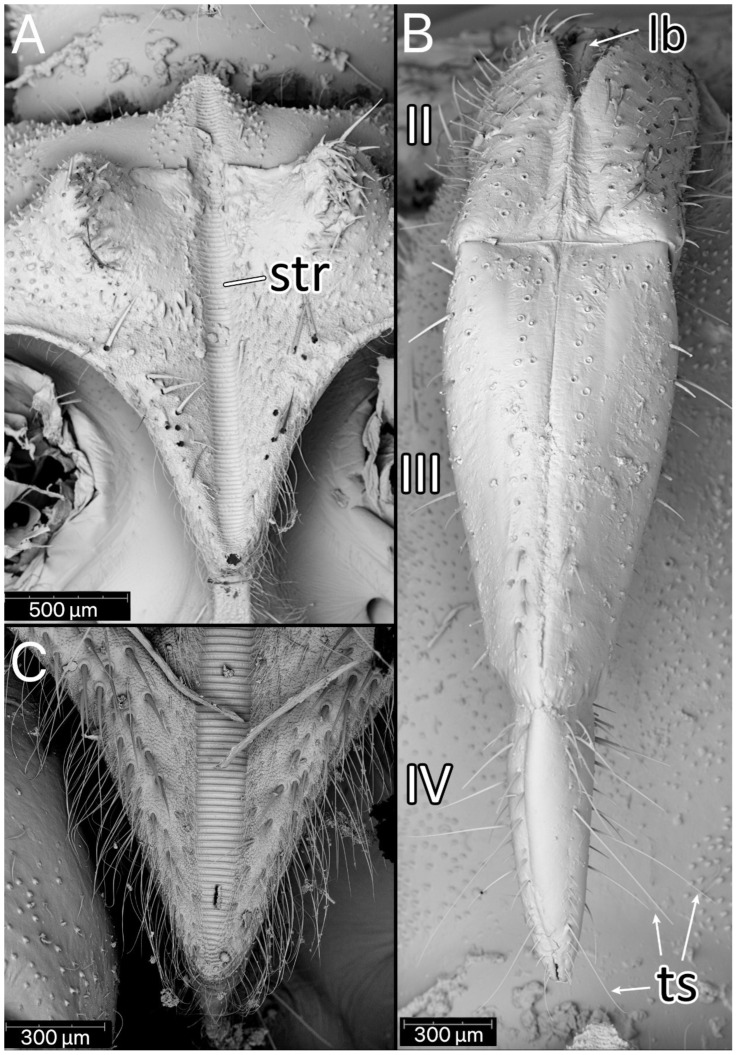
SEM views. (**A**) Elongated and narrow stridulitrum of the fifth instar of *Psyttala horrida* (Stål, 1865) with distinct bulges on both sides; (**B**) details of visible labial segments with elongated *sensilla trichodea*, fifth instar; (**C**) tip of the stridulitrum with details about density and types of setae, fourth instar. **lb**—labrum; **str**—stridulitrum; **ts**—*sensillum trichodeum*; **II**–**IV**—labial segments.

**Figure 20 insects-13-01014-f020:**
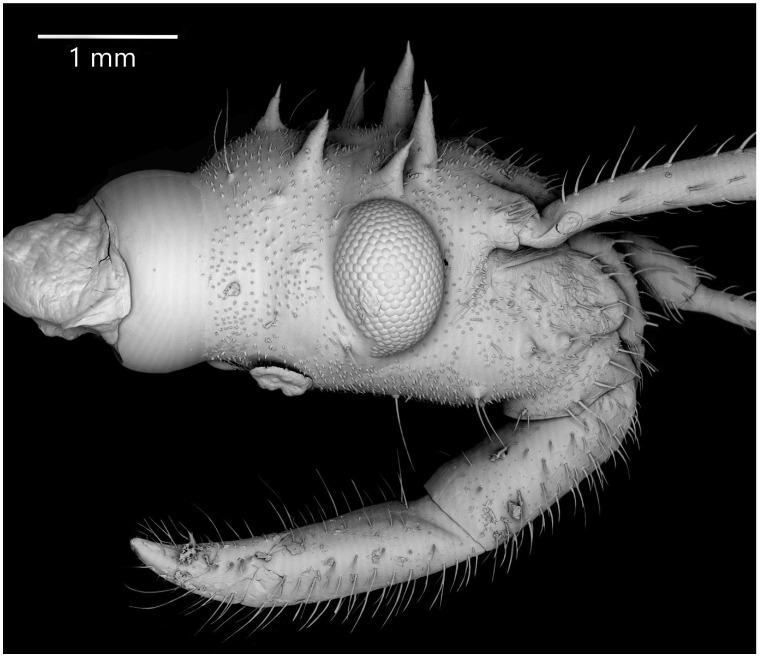
*Psyttala horrida* (Stål, 1865), third instar, head in lateral view, SEM.

**Figure 21 insects-13-01014-f021:**
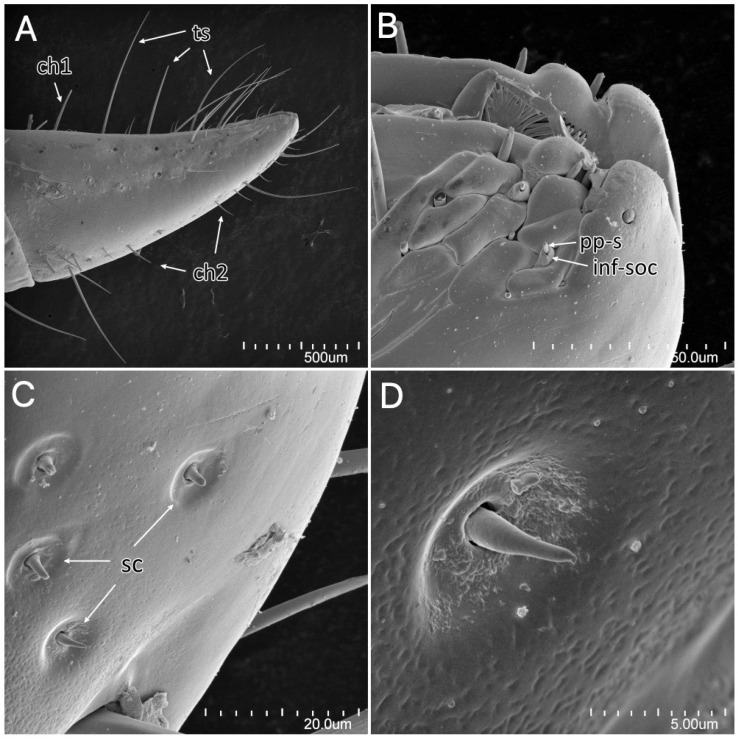
*Psyttala horrida* (Stål, 1865) third instar, SEM views. (**A**) Fourth visible labial segment with *sensilla chaetica* and *trichodea*; (**B**) sensory field at the tip of the labium; (**C**) *sensilla coeloconica* on the distal part of the labium, right under the sensory field; (**D**) magnification of the single *sensillum coeloconicum*. **ch1**—*sensillum chaeticum* type 1; **ch2**—*sensillum chaeticum* type 2; **inf-soc**—inflexible socket; **pps**—porus peg sensillum; **sc**—*sensillum coeloconicum*; **ts**—*sensillum trichodeum*.

**Figure 22 insects-13-01014-f022:**
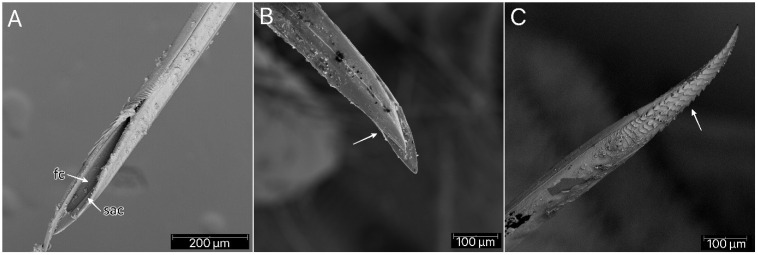
*Psyttala horrida* (Stål, 1865) fourth instar, SEM views. (**A**) Food and salivary canals of the maxillary stylets; (**B**) maxillary stylet with ridges on the interior surface; (**C**) ridges with hooklets at the apex of the mandibular stylet. **fc**—food canal; **sac**—salivary canal.

**Figure 23 insects-13-01014-f023:**
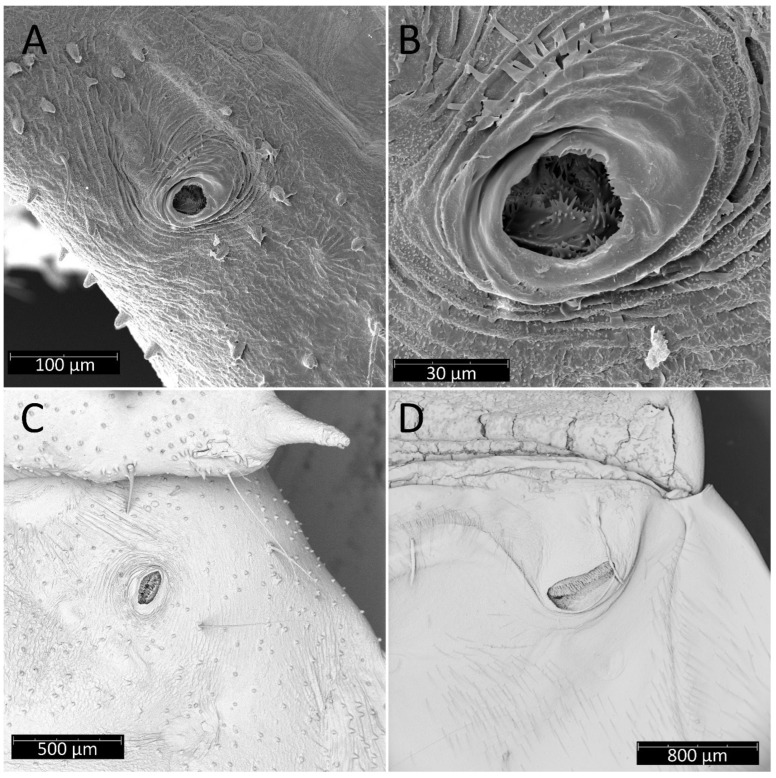
Spiracles on the dorsal side of the first abdominal segment of *Psyttala horrida* (Stål, 1865): (**A**) second instar; (**B**) magnification of the interior; (**C**) fourth instar; (**D**) adult.

**Figure 24 insects-13-01014-f024:**
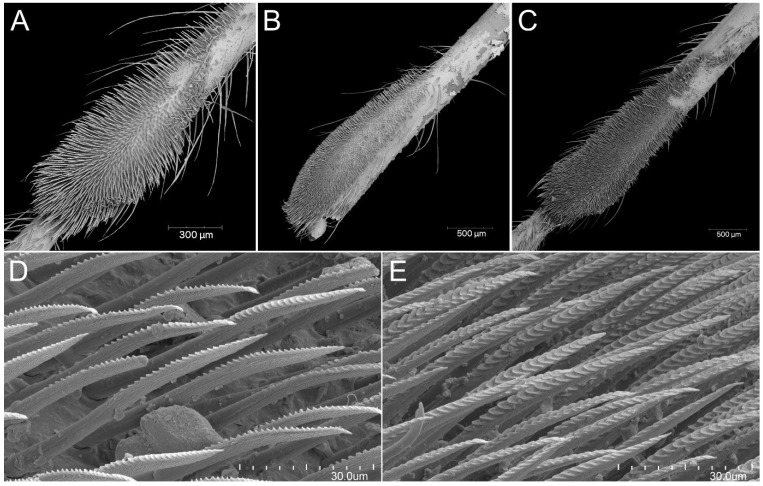
*Psyttala horrida* (Stål, 1865), a pad of barbed setae on the ventral apical part of foretibia of (**A**) second, (**B**) third and (**C**) fourth instar; (**D**,**E**) magnification of barbed setae from both sides.

**Figure 25 insects-13-01014-f025:**
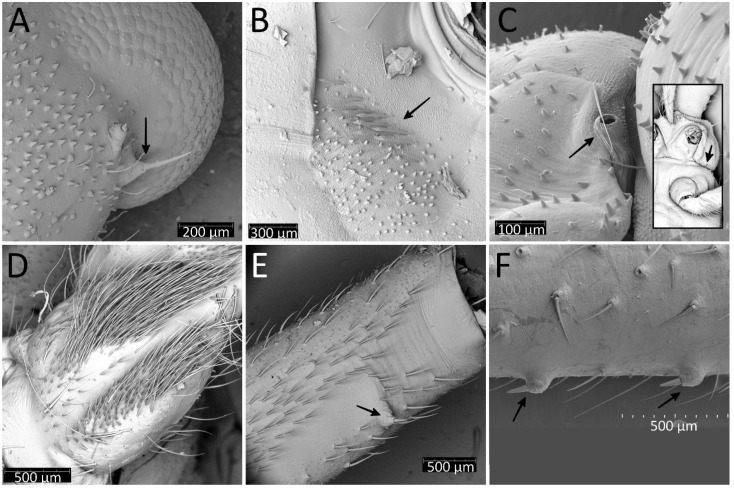
*Psyttala horrida* (Stål, 1865), fourth instar, SEM views. (**A**) An appendage near the eye on the ventral side; (**B**) pointed appendages visible under the forecoxa; (**C**) a spiracle-like structure at the lateral edge of the propleuron, visible only in the first and second instar; (**D**) two stripes with dense, long setae on the middle trochanter, ventral view; (**E**) an appendage on the distal part of the middle femur, ventral view; (**F**) appendages on the middle part of the hind femur, lateral view.

## Data Availability

The data presented in this study are available in the article.

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
