# Peer review of "Psyttala horrida (Stål, 1865) (Hemiptera: Reduviidae: Reduviinae)—A Morphological Study of Eggs and Nymphs"

_insects, 2022, doi:10.3390/insects13111014_

Round 1
Reviewer 1 Report
General comments
1. The manuscript is well-written, but I suggest the text be sent for revision by a native English speaker. There are many grammatical errors, but due to time constraints, I have corrected only a tiny amount of these errors.
2. Moreover, I have some significant reservations about this MS. Therefore, I recommend major revisions before the text can be accepted for publication.
3. The authors should, at least shortly, discuss their results in light of previous studies on eggs and instars morphology in Reduviidae (the chapter “Discussion” should be added).
Specific comments
1. Introduction
- A monograph by Cobben (1968) is not included [Evolutionary trends in Heteroptera. Part I. Eggs, architecture of the shell, gross embryology and eclosion]. The authors should consider Cobben’s data on Reduviidae in the introductory part and in the discussion.
- 3.2. Egg structure: the authors wrote, “No holes have been identified on the chorion surface, neither micropyles nor aeropyles”. However, to my knowledge, they are present (see also Cobben 1968). SEM studies are inappropriate for observing micropyles and aeropyles; they are better visible when the chorion is studied using a light microscope technique.
2. Material and Methods
- The authors should explicitly explain and cite the papers on which the morphological nomenclature of instars and eggs was based. The morphological nomenclature is sometimes a little misleading in the present text form.
3. “Taxonomic accounts” should be changed to Results
- It is hard to understand the homology of some structures during ontogeny (first to the fifth instar), e.g., “the spine-like process/setae”. It would be necessary to provide a table (or a simple graphical explanation) in which all data related to this structure are summarized (I, II, III transversal row in pro-, meso- and metapleuron, for all instars). It would help to understand how this structure changes during the species ontogeny.
[4.] Discussion (should be added to the text)
- The authors should discuss their results in light of previous data on eggs and instars morphology in Reduviidae (especially those provided by Cobben, 1968).
4. [5.] Conclusions
- Why the presence of the spiracle on pronotum in the first instar was mentioned only in “Conclusion”. It should be described in the “Results” section. Moreover, the presence of a spiracle on the pronotum in the 1st instar is very unusual in Heteroptera, because the spiracles were recorded usually only on meso- and metathorax. Can the authors explain this phenomenon – maybe someone has already noticed this pronotal spiracle? If not, it should be mentioned in the Discussion.
All other detailed comments were put directly into the text (the Word file).

Author Response
Thank you for reviewing our work and drawing attention to some important issues. Your comments were helpful, and we followed them. You wrote: "3.2. Egg structure: the authors wrote, "No holes have been identified on the chorion surface, neither micropyles nor aeropyles". However, to my knowledge, they are present (see also Cobben 1968). SEM studies are inappropriate for observing micropyles and aeropyles; they are better visible when the chorion is studied using a light microscope technique." We also used the light microscopy method, but we still haven't been able to identify the micropyle on the chorion. However, we did find a valuable comment from Cobben that may apply to the eggs we examined. He wrote: "Absence of micropyles can be explained where the eggs are fertilized early in the ovarioles before the chorion is deposited {e.g. Metatetranychus, BEAMENT, 1951). This is found also in Heteroptera with traumatic insemination where the number of micropyles is reduced from two, through one, to zero in several Cimicoidea." Thus, your suggestion to include this article in our manuscript was very helpful.
Despite our best efforts, we could not find any mention of propleuron spiracle in Heteroptera nymphs. However, it is possible that this is not a rare structure, but is rarely found. Locating it requires observation from the underside.
As recommended, we supplemented the article with citations about terminology and added more information in the "material and methods" section. In addition, we changed the concept of the article and supplemented it with a "discussion" section.
Reviewer 2 Report
Dear Authors,
The article presented for review describes the morphology of nymphs and eggs. The descriptions themselves are very detailed and mostly correct. However, the nomenclature of sensory structures requires re-examination and improvement.
On the other hand, several aspects of the article need to be rethought.
All detailed comments have been placed directly in the text.
Below are some general comments that may help to improve the article.
- references concerning taxonomy should be supplemented,
- method descriptions should be completed (nomenclature issue),
- consider the layout of individual sections; the current headers do not match the content. Currently, the "conclusions" (at the moment, this section contains the results.) or the "discussion" section is entirely missing. 60 out of 68 references were mentioned when listing species that have already been studied (in the introduction). The results obtained were not embedded in any context. It is worth discussing issues such as: have structures similar to those observed been found in other species? What are their supposed functions? Has anything of particular interest been observed? Does any aspect require further research?
Best wishes

Author Response
Thank you for reviewing our work and drawing attention to some important issues. We fully agree with the reviewer's comment regarding the citation of publications with the described taxa. The publications mentioned in the text were included in the list of references. We provide the author's names and data for all mentioned species; however, citations are provided only for the species under investigation and for the higher taxa to which it belongs. We know that the citation method in Insects does not disturb the structure of the text, but in our opinion, adding a double citation to a species description work and then to a work where eggs/nymphs are described is confusing.
As recommended, we supplemented the article with citations about terminology and added more information in the "material and methods" section. In addition, we changed the concept of the article and supplemented it with a "discussion" section.
Other comments we referred to:
- following the reviewer's remark, we changed some keywords
- 'bothrium with trichobothrium' – we understand the correctness of the reviewer's remark and changed the description to a more appropriate one
- we asked someone who specializes in sensillae, and she confirmed it would be sensillum coeloconicum – thank you for pointing this out
Reviewer 3 Report
Please find comments in attached file. If possible the chresonymy section should be updated. Check the language and use British or American style consecutively. CHeck the references and correct according to unified format and journal style.

Author Response
Thank you for reviewing our work and drawing attention to some important issues. First, we prepared the list of the misspelt names in the species taxonomy description. As mentioned by the reviewer, 'to avoid further confusion and stabilize the name'.
Other comments we referred to:
- of course, the correct abbreviation is spp. because we mean a group of species from the given genus and not a subspecies. This is our spelling mistake, and we want to thank the reviewer very much for noticing it; otherwise, it would change the meaning of our statement
- to our knowledge resulting from the analysis of English dictionaries (for example, Collins), the word 'colouration' is correct in British English, while 'coloration' is correct in American English. The rest of the reviewers seem to agree, as they did not indicate the spelling of the word as a mistake
- we removed the duplicate numbering from the references and adjusted the titles according to the unified format
- other issues, such as word breaks or the positioning of photo captions and the photos themselves, result from the journal's layout. We will pay attention to it if the article is accepted so that everything in the received proof is legible and adjusted
Reviewer 4 Report
Dear authors, I have made constructive comments in the annex. Congratulations on the work.

Author Response
Thank you for reviewing our work and drawing attention to some important issues. We supplemented all species and genera names with the authors and the year of description. However, citations are provided only for the species under investigation and the higher taxa to which it belongs, as suggested by another reviewer. We have completed the information about the colony. Unfortunately, we do not have the appropriate photo to show it in publication. However, we will remember it in the future. We have completed the remaining descriptions in the "materials and methods" section. We improved the references under the figures and supplemented the article with a "discussion" section.
Round 2
Reviewer 1 Report
I have detected only two small mistakes that needed corrections.
First, please correct the name "Piratinae" to "Peiratinae" in "Discussion".
Moreover, as I know now which seta on Fig. 16 A-C was named "bifurcated" I suggest naming it "cirrose" or "cirrous" (according to Torre-Bueno's "A glossary of entomology"). Or you can find another more adequate name, because a bifurcated seta is a seta with two branches of equal or nearly equal length arising together, usually below the distal third, commonly at the base. The seta in Fig. 16 A-C does not match this definition.
Author Response
Dear Reviewer,
Thank you very much for your comments, which were very helpful for us. The name of Peiratine was changed by autocorrect, which we did not notice. Thank you very much for noticing this. The hair structure definition has also been changed to cirrous according to your suggestion. This word describes its structure better. Thank you.
Reviewer 2 Report
Dear Authors,
The revised paper is ready for publication.
Best wishes
Author Response
Dear reviewer,
Thank you for your comments and for helping us improve the manuscript.